# Epidemiology and Genetic Diversity of *Spirometra* Tapeworm Isolates from Snakes in Hunan Province, China

**DOI:** 10.3390/ani12091216

**Published:** 2022-05-09

**Authors:** Tengfang Gong, Xiaoyi Su, Fen Li, Junlin He, Shuyu Chen, Wenchao Li, Xinrui Xie, Yisong Liu, Xi Zhang, Wei Liu

**Affiliations:** 1Research Center for Parasites & Vectors, College of Veterinary Medicine, Hunan Agricultural University, Changsha 410128, China; gongtf@stu.hunau.edu.cn (T.G.); loislf@163.com (F.L.); hejunlin607@163.com (J.H.); shuyuchen2021@stu.hunau.edu.cn (S.C.); leo0725@stu.hunau.edu.cn (W.L.); 1981318528@stu.hunau.edu.cn (X.X.); liuyisong@hunau.edu.cn (Y.L.); 2Department of Parasitology, School of Basic Medical Sciences, Zhengzhou University, Zhengzhou 450001, China; suxy836@163.com; 3The Key Laboratory of Animal Vaccine & Protein Engineering, Changsha 410128, China

**Keywords:** *Spirometra* tapeworm, sparganum, snake, prevalence, genetic diversity

## Abstract

**Simple Summary:**

With this study, we aimed to investigate the epidemiology and genetic diversity of *Spirometra* tapeworms in snakes in Hunan province. The result showed that the positivity rate among snakes was 89.53%, which is the highest among other regions. Genetic diversity analysis based on concatenated sequences revealed high genetic diversity but no distinct genetic structure among *Spirometra* populations. Phylogenetic analysis supported the division of European and Chinese *Spirometra* isolates and a single species in Chinese *Spirometra* isolates.

**Abstract:**

Sparganosis, caused by the plerocercoid larvae of *Spirometra* tapeworms, is a public health hazard worldwide. The prevalence and genetics of sparganum from snakes remain unclear. In this study, we investigated the prevalence of sparganum infection in wild snakes in Hunan province and compared the prevalence of *Spirometra* tapeworms in snakes worldwide. Furthermore, the genetic diversity of collected isolates was analyzed using mitochondrial *cyt*b and *cox*1 genes. The result shows that the sparganum infection rate in wild snakes (89.50%, 402/449) was higher in Hunan than in other regions. Genetic diversity analysis based on concatenated sequences revealed high genetic diversity but no distinct genetic structure among *Spirometra* populations. Phylogenetic analysis supported the division of European and Chinese *Spirometra* isolates and a single species in Chinese *Spirometra* isolates. The prevalence of *Spirometra* tapeworms in snakes is serious, and the risk of sparganosis should be further publicized.

## 1. Introduction

The plerocercoid larvae (sparganum) of *Spirometra* tapeworms can parasitize humans and cause an important foodborne parasitic zoonosis known as sparganosis [1,2]. Sparganosis occurs worldwide, especially in eastern and southeastern Asian countries [3]. More than 1300 cases of sparganosis have been reported in China, and the actual number of infections may be far higher because many cases may not be recognized or reported [4].

As the most common intermediate hosts in the life cycle of *Spirometra* tapeworms, snakes and frogs transmit sparganosis to humans in China [5,6]. Humans can be infected by consuming raw or undercooked snake/frog meat or by using raw snake/frog flesh in traditional poultices [3]. Sparganum has been extensively isolated from frogs in different regions of China [6]. However, as a very important host, information on the prevalence of sparganosis in snakes remains scarce [7,8,9]. Therefore, knowledge regarding the prevalence of sparganum infection in snakes is valuable for preventing and controlling sparganosis in humans.

By using mitochondrial genes or the complete mitochondrial genome the sparganum isolates collected in different China locations were classified as *S. erinaceieuropaei* [6,10,11,12]. No other species of *Spirometra* has been reported as a source of human infection in China. However, two *Spirometra* species, *S. erinaceieuropaei* and *S. decipiens*, collected from snakes (*Dinodon rufozonatum* and *Agkistrodon saxatilis*) have been identified as species in China through morphological and genetic methods [7]. Therefore, the precise identification of *Spirometra* species from snakes in China requires further investigation.

Data on *Spirometra* tapeworm infection of snakes in Hunan province were supplemented for the first time in this study. Meanwhile, the prevalence of *Spirometra* tapeworm in snakes worldwide was also analyzed. Two validated markers, mitochondrial cytochrome B (*cyt*b) and cytochrome C oxidase subunit I (*cox*1), which have been verified as suitable markers for inferring genetic population differences of *Spirometra* tapeworms [11,13,14,15], were sequenced and analyzed to infer an exhaustive genetic diversity analysis of *Spirometra* tapeworms in snakes from different locations in Hunan province.

## 2. Materials and Methods

### 2.1. Collection of Sparganum Isolates from Snakes

Samples were collected from field sites in 15 geographical locations of Hunan province in China from April 2018 to October 2019 (Figure 1). The presence of sparganum was examined according to the method described by Liu et al. [5]. In brief, snakes were euthanized using ethyl-ether anesthesia and skinned. The skin was peeled off from the neck to the tip of the tail, and the visceral mass was measured from the esophagus and trachea to the cloaca. Then, the number of spargana was counted. In addition, all surveys of sparganum infection in snakes from other publications were included the analysis. All collected worms were fixed in 99% ethanol and kept at −20 °C for molecular analysis.

### 2.2. Sequencing of Target Genes

Genomic DNA was extracted using a Wizard^®^ SV genomic DNA purification system (Promega, Madison, WI, USA) according to the manufacturer’s protocol. The expression of two mitochondrial markers (*cox*1 and *cyt*b) was analyzed using the primer combinations listed in Appendix A. PCR products were purified using an EasyPure PCR purification kit (Transgen, Beijing, China) and sequenced in both directions by Tsingke Company (Beijing, China).

### 2.3. Genetic Diversity Analysis of Isolates from Snakes

The DNA sequences of *cyt*b and *cox*1 were initially aligned using the program Clustal X v.2.0 [16] and adjusted in MEGA v.7.0 according to their amino acid sequences [17]. The nucleotide composition, conserved sites, variable sites, parsimony-informative sites, and singleton sites were estimated using MEGA v.7.0. The haplotypes were inferred, and genetic diversity values per population were calculated using DnaSP v.6 [18]. The median-joining network representing the relationship among all obtained haplotypes was prepared using PopART v.1.7 [19]. Analysis of molecular variance was completed using Arlequin v.3.5.1 to detect the partitions of genetic diversity within and among populations [20]. Pairwise genetic distances were also estimated using Arlequin v.3.5.1 to explore levels of genetic differentiation among the populations. To test the demographic change of the *Spirometra* population, we performed neutrality tests using Arlequin v.3.5.1 with mismatch distribution using the sum of squared deviations and raggedness index (RI) between observed and expected mismatches and Fu’s FS test and Tajima’s D [21,22].

### 2.4. Phylogenetic Analysis

All available complete sequences of *cox*1 and *cyt*b in the GenBank database were included to perform a phylogenetic analysis of *Spirometra* tapeworms (Appendix A). Four *Dibothriocephalus* species-*D. nihonkaiense* (Genbank accession number representing the *cyt*b/*cox*1 genes: AB508837/AB015755), *D. latum* (AB522608/AB511963), *D. dendriticum* (AB522613/KC812045), and *D. ditremum* (AB522617/FM209182)-were used as the outgroup. The phylogeny of all collected sequences was estimated by the maximum likelihood (ML) and maximum parsimony (MP) methods. ML analysis was performed in PhyML v.3.0 [23] using models selected by jModelTest 2 under the Akaike information criterion [24]. The support of each internal branch of the phylogeny was estimated using non-parametric bootstrapping (1000 replicates). MP analyses were performed in PAUP*4b10 using heuristic searches with TBR branch swapping and 10,000 random addition sequences. The confidence of each node was assessed by bootstrapping (2000 pseudo-replicates and heuristic search of 20 random addition replicates with TBR option).

## 3. Results

### 3.1. Prevalence of Sparganum Infection in Snakes

A total of 2934 snakes belonging to 28 species were surveyed for sparganum infection. As a result, 1581 (53.89%, 1581/2934) were found to be positive (Table 1). The level of sparganum infection in wild snakes in Korea was the highest (83.04%, 235/283), followed by China (51.93%, 1152/2218) and Indonesia (50.85%, 192/378). In contrast, the infection rate among snakes from Poland was only 3.64%, which is far lower than that among snakes from Asian countries. In China, the highest prevalence was found in Zhejiang province (100%, 5/5), followed by Shanghai (93.22%, 55/59) and Hunan (89.53%, 402/449). In central China’s Hunan province, 449 wild snakes were investigated in 14 representative regions in this study. The prevalence ranged from 65% to 100%, with an infection intensity of 1-70 spargana per snake. The highest infection rate was found in snakes from Shaoyang (100%, 20/20), followed by Zhangjiajie (96.67%, 29/30) and Xiangtan (95.77%, 68/71). Generally, the infection rate in Hunan province is higher than that in other provinces in China, such as Guangdong (44.51%, 353/757), Fujian (78.79%, 25/32), Guangxi (23.17%, 38/160), Guizhou (42.16%, 85/172), Jinlin (30.80%, 134/435), and Hebei (36.90%, 55/149).

Among all 28 collected snake species, only 5 species, namely *Naja kaouthia*, *Enhydris bocourti*, *Elaphe mandarinus*, *E. schrenkii*, and *E. davidi*, showed no sparganum infection, indicating that these snake species were probably insensitive to sparganum infection (Table 2). The highest infection rate was found in *Zaocys dhumnades* (94.24%, 393/417), followed by *Dinodon rufozonatum* (86.89%, 126/145) and *Agkistrodon saxatilis* (85%, 51/60). Interestingly, the highest infection intensity of sparganum was also found in the *Z. dhumnades* with infection intensity of 1-294 spargana per snake, followed by *D. rufozonatum*, with an infection intensity of 1-291 spargana per snake; *Ptyas mucosus*, with an infection intensity of 1-208 spargana per snake; and *Naja atra*, with an infection intensity of 1-208 spargana per snake.

### 3.2. Genetic Diversity and Phylogenetic Pattern

All amplifications for 67 sparganum isolates were successful, with 1110-bp PCR products for *cyt*b and 1566-bp products for *cox1*. These sequences detected 119 polymorphic sites (61 for *cyt*b and 58 for *cox*1), with 83 parsimony-informative sites (39 for *cyt*b and 44 for *cox1*) and 36 singleton sites (22 for *cyt*b and 14 for *cox1*). The concatenated sequences identified 48 haplotypes within the 67 isolates, originating from 15 localities. Both individual and combined sequences for *cox1* and *cyt*b had high Hd, accompanied by low Pi (Table 3), which is consistent with previous analyses of sparganum isolates from frogs from different locations in China and isolates from different hosts in Poland [13,15].

Analysis of molecular variance indicated that most of the observed genetic variation occurred within the 14 endemic populations (70.57%), whereas the difference among the populations contributed 24.96% to the total population (Table 4). The pairwise fixation index (FST) values between specified regions were estimated to measure the population differentiation (Table 5). Across all estimated 91 pairwise FST values, only 22 exhibited statistical significance. Within these 22 statistically significant FST values, most FST values between CS and other endemic regions were above 0.25. These findings indicate high genetic differences between isolates from CS and isolates from other geographical regions in Hunan province. One reason for this observation might be that samples from CS were isolated from four different host species (*Z**. dhumnades*, *Panthera tigris*, *Prionailurus bengalensis*, and *Felis silvestris*, Cat).

Analysis of the concatenated sequences showed no distinct genetic structure across the sampled *Spirometra* populations in Hunan province. In the median-joining network, all 67 sequences united in a star-like shape (Figure 2). Although the haplotypes were high (48 haplotypes), no segregation was detected by population, region, or host species. Haplotype 26 was the most prominent, represented by eight samples from two regions (CS and ZZ) and isolated from all five host species. Neutrality tests of Tajima’s D and Fu’s FS for the whole Hunan dataset showed a significant negative value of Fu’s FS (FS = −7.7694, *p* = 0.043) but a non-significant positive value of Tajima’s D (D = 0.6439, *p* = 0.798). Mismatch distribution analyses revealed multimodal frequency distributions, which did not support a demographic expansion of *Spirometra* populations in Hunan province (Figure 3). In addition, low values of the sum of squared deviation and RI under the demographic expansion model were found. The results of Bayesian skyline plot analyses also rejected sudden population expansion.

### 3.3. Phylogenetic Pattern

In this study, we collected 67 sparganum isolates from 15 geographical locations in Hunan province to explore the phylogenetic diversity of *Spirometra* isolates and compare the genetic variance between the isolates collected in frogs from other geographical locations. The basic identical tree topologies were generated through phylogenetic inference based on ML and MP methods. The phylogenetic pattern based on the MP analysis is shown in Figure 4. All collected isolates were grouped into two distinct clades (Clade 1 and Clade 2) with high support values (bootstrap values = 100). Clade 1 and Clade 2 were isolates from Poland and China, respectively. However, no clear phylogenetic structure was found within both clades. Within Chinese isolates (Clade 2), although several isolates in snakes and frogs were grouped to form subclades; the support values of these subclades were too low to confirm the phylogenetic patterns, indicating that the Chinese isolates should be considered a single species.

## 4. Discussion

China is home to the most sparganosis, accounting for 80% of the worldwide population. Sparganosis infection generally occurs in humans as a result of eating raw or undercooked frogs and snakes with sparganum [3]. In this study, we conducted a large-scale survey of sparganum infection in wild snakes from 14 geographical locations in Hunan province to understand the prevalence of *Spirometra* tapeworms in wild snakes. Our results showed that the average prevalence of sparganum infection in snakes in Hunan province was higher than that in Guangdong, Fujian, Guangxi, Guizhou, Jinlin, and Hebei and lower than that in Zhejiang and Shanghai. Unsurprisingly, snake infection rates are higher than frog infection rates in Hunan province (20.20%, 59/292) [12]. As a paratenic host in the life cycle of *Spirometra* tapeworms, snakes can enrich sparganum by preying on other infected animals, such as frogs. The prevalence of *Spirometra* tapeworms in snakes worldwide is generally high, suggesting that sparganum infection in snakes may be an important mode of transmission. A total of 28 species of snakes were surveyed, and *Z. dhumnades* showed the highest infection rate and intensity, which might be because *Z. dhumnades* is a large species and can therefore be parasitized by more sparganum. In the south and southwest regions of China, snakes are considered a delicacy. Notably, this study showed that the infection rate of *Z. dhumnades* and *E. taeniura*, the main edible snakes in China, is high.

The distinct genetic separation of the Polish and Chinese populations of *Spirometra* isolates was identified by Kołodziej-Sobocińska, who suggested that Polish and Chinese should be two species [15]. Using a global full-length DNA sequence dataset of *cox*1, Kuchta performed a comprehensive phylogenetic analysis of *Spirometra* tapeworms [14] and suggested that there are at least six distinct lineages of the genus:*S. mansoni* lineage (corresponding to most of the isolates from Asia, Australia, Romania, and a single sequence from Tanzania);*Spiro**metra* sp. 1 lineage (corresponding to a few specimens from Korea and Japan);*Spirometra folium* lineage (corresponding to most specimens from Africa, such as Sudan and Ethiopia and the remaining specimens from Tanzania);*S. erinaceieuropaei* lineage (corresponding to the majority of European specimens);*S. decipiens* complex 1 lineage; and*S. decipiens* complex 2 lineages.

Yamasaki re-examined *Spirometra* samples from Asia based on *cox*1 DNA sequence data and suggested two distinct *Spirometra* species [35]. Type I is genetically diverse and widely distributed; however, Type II has only been found in Japan and Korea. In this study, phylogenetic analysis supported the division of Polish and Chinese *Spirometra* isolates, which supports the findings of Kołodziej-Sobocińska et al. and Kuchta et al. Regarding Chinese *Spirometra* isolates, Zhang explored the genetic diversity of sparganum isolates in frogs from eastern, central, southern, and southwest China [36]. They found that the Chinese *Spirometra* population probably comprises two subgroups. However, in this study, we found no distinct genetic structure of *cyt*b and *cox*1 genes among *Spirometra* populations in snakes from Hunan province. One possible reason could be that the geographic distances between populations studied here and the sampling size were relatively small. In addition, the results also show that all isolates in snakes and frogs belong to one species, which provides support for a single species in Chinese *Spirometra* isolates. Until now, definitive morphological criteria for distinguishing *Spirometra* species, including sparganum and adult worms, have not been established [35]. Therefore, the taxonomy of *Spirometra* and their phylogenetic relationships remain ambiguous, and more morphological and molecular studies are warranted to clarify the systematics of the genus.

## 5. Conclusions

The survey results showed that sparganum infection rates in wild snakes in Asian countries were higher than in Europe. Most snake species (82.14%, 23/28) were sensitive to sparganum infection, and the most frequently infected species was *Z. dhumnades*, followed by *D. rufozonatum* and *A. saxatilis*. The sparganum infection rates in wild snakes in several regions of China were still high, especially in Zhejiang, Shanghai, and Hunan. Genetic diversity analysis based on *cyt*b and *cox*1 genes revealed no distinct genetic structure among *Spirometra* populations in Hunan province. Phylogenetic analysis supported the division of European and Chinese *Spirometra* isolates and a single species in Chinese *Spirometra* isolates.

## Figures and Tables

**Figure 1 animals-12-01216-f001:**
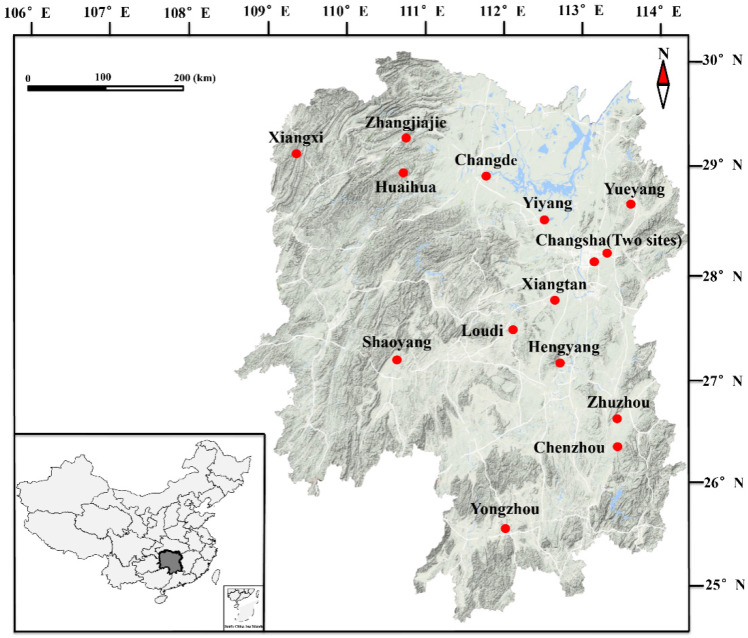
Sampling sites in Hunan province, China. The geographic location is shown in the inset. The sampling sites were added according to GPS data.

**Figure 2 animals-12-01216-f002:**
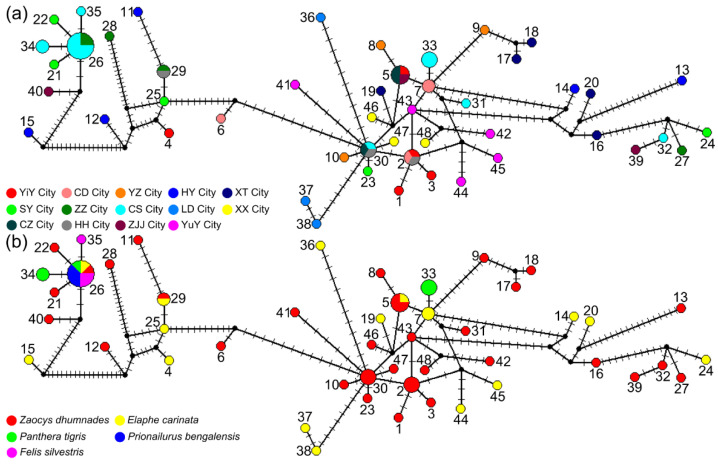
Median-joining network of 67 sequences of *cyt*b and *cox*1 genes in sparganum isolates colored by sampling sites (**a**) and host species (**b**) in Hunan province, China. The area of circles represents the number of individuals with that haplotype. Perpendicular short lines on the branches indicate unsampled intermediate haplotypes.

**Figure 3 animals-12-01216-f003:**
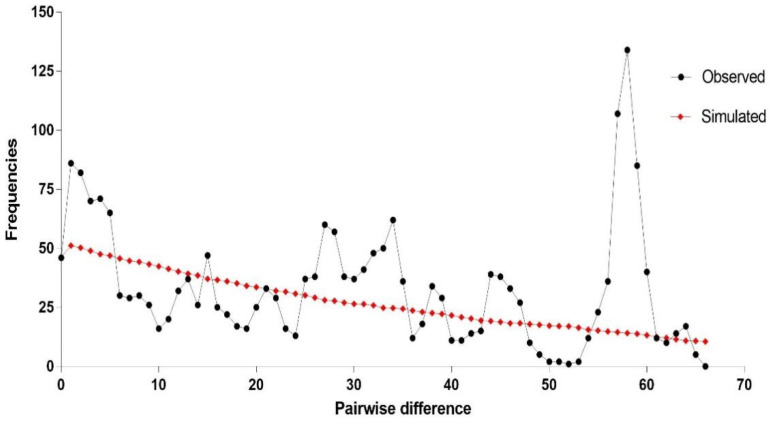
Estimate of demographic expansion of sparganum isolates from Hunan province. (a) Mismatch distribution analyses. The line charts represent the observed frequencies of pairwise differences among haplotypes. (b) Bayesian skyline plot calculations. The *X*-axis is in units of million years in the past, and the *Y*-axis is Ne × μ (effective population size × mutation rate per site per generation). The median estimates are shown as thick solid lines, and the 95% HPD limits are represented by the colored areas.

**Figure 4 animals-12-01216-f004:**
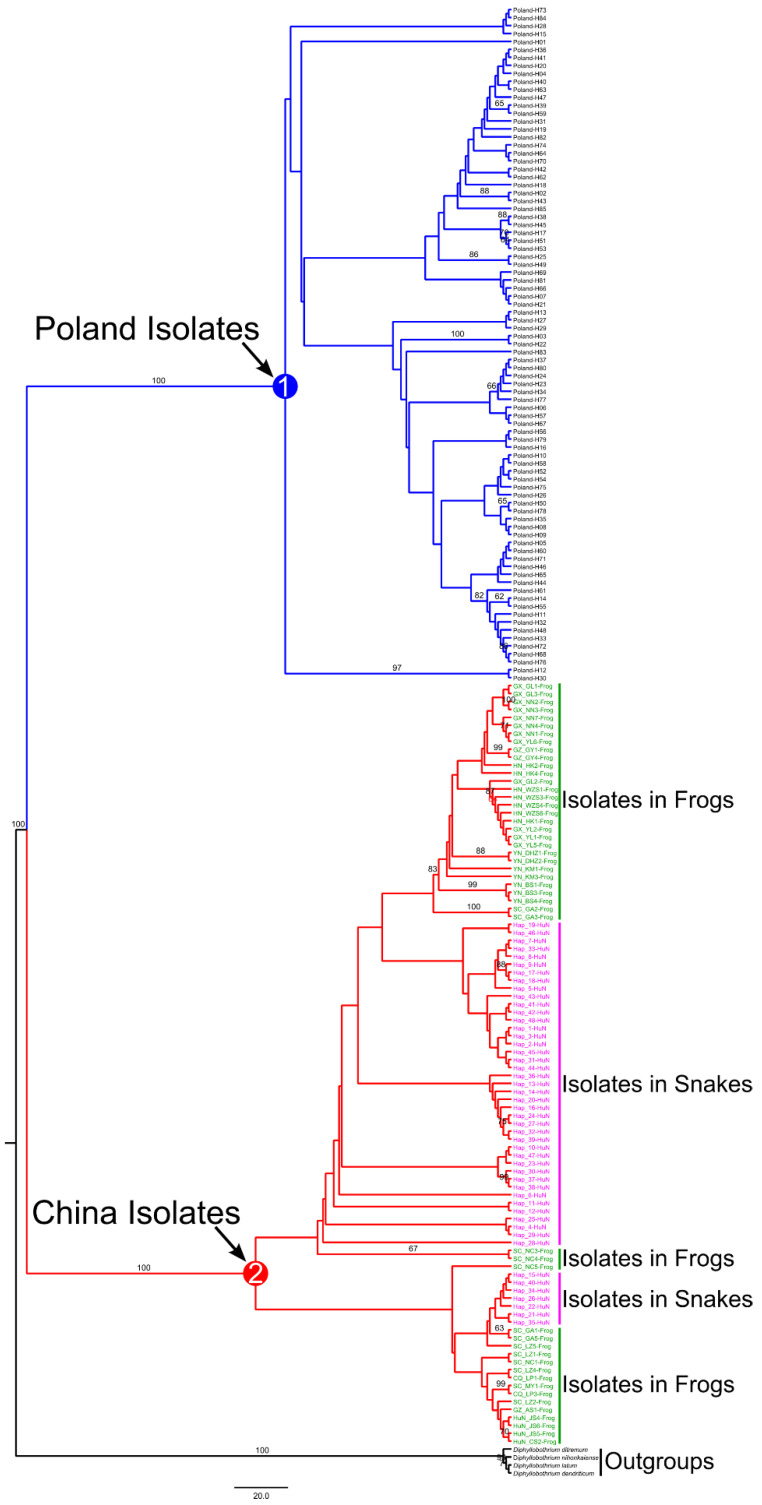
Phylogenetic relationships among the examined sparganum isolates from different locations in China and Poland were inferred by maximum parsimony (MP) analysis based on the concatenated sequences of *cyt*b and *cox*1. The numbers along branches indicate bootstrap values, and bootstrap values above 60 are shown. Circled numbers represent the main clades discussed in the text.

**Table 1 animals-12-01216-t001:** Prevalence of sparganum infection in snakes.

Country	Geographical Origin	No. Infected/No. Examined (%)	Infection Intensity (Spargana/Snake)	References
China				
Hunan	HN-YiY	43/49 (87.76)	5-56	This study
	HN-CD	18/20 (90.00)	5-25	This study
	HN-YZ	28/33 (84.85)	1-18	This study
	HN-HY	13/20 (65.00)	4-22	This study
	HN-XT	68/71 (95.77)	5-70	This study
	HN-SY	20/20 (100.00)	5-60	This study
	HN-ZZ	22/23 (95.65)	2-32	This study
	HN-CS	60/71 (84.51)	3-60	This study
	HN-LD	22/24 (91.67)	5-49	This study
	HN-CZ	18/20 (90.00)	7-25	This study
	HN-HH	18/20 (90.00)	6-35	This study
	HN-ZJJ	29/30 (96.67)	8-38	This study
	HN-YuY	24/28 (85.71)	2-22	This study
	HN-XX	19/20 (95.00)	8-43	This study
	Total	402/449 (89.53)	1-70	
Guangdong		353/757 (44.51)	1-213	[25,26,27]
Fujian		25/32 (78.79)	1-28	[28]
Guangxi		38/160 (23.17)	1-208	[29,30]
Guizhou		85/172 (42.16)	1-121	[31]
Jilin		134/435 (30.80)	N/a	[32]
Zhejiang		5/5 (100.00)	2-99	[33]
Shanghai		55/59 (93.22)	1-294	[34]
Hubei		55/149 (36.90)	1-23	[8]
Total		1152/2218 (51.93)	1-294	
Korea		235/283 (83.04)	N/a	[7]
Indonesia		192/378 (50.85)	1-111	[9]
Poland		2/55 (3.64)	1-3	[15]
Total		1581/2934 (53.89)	1-294	

**Table 2 animals-12-01216-t002:** Prevalence of sparganum infection in different snake species.

Species	No. Infected/No. Examined	Prevalence (%)	Infection Intensity (Spargana/Snake)
*Deinagkistrodon acutus*	5/11	45.45	1-34
*Bungarus multicinctus*	6/49	12.24	1-26
*Bungarus fasciatus*	1/10	10.00	1-1
*Naja atra*	46/184	25.00	1-208
*Naja naja*	9/16	56.25	1-28
*Naja kaouthia*	0/7	0	N/a
*Dinodon rufozonatum*	126/145	86.89	1-291
*Elaphe carinata*	170/216	78.70	1-172
*Elaphe taeniura*	70/150	46.67	1-121
*Coelognathus radiatus*	3/15	20.00	1-4
*Enhydris bocourti*	0/9	0	N/a
*Enhydris chinensis*	3/35	8.57	N/a
*Enhydris plumbea*	11/95	11.58	1-28
*Ptyas korros*	47/127	37.01	1-65
*Ptyas mucosus*	67/150	44.67	1-208
*Xenochrophis flavipunctatus*	86/132	65.15	1-133
*Zaocys dhumnades*	393/417	94.24	1-294
*Sinonatrix annularis*	11/38	28.95	1-14
*Cryptelytrops albolabris*	3/5	60.00	1-8
*Natrix natrixLinnaeus*	5/65	7.69	1-14
*Elaphe mandarinus*	0/2	0	1-26
*Gloydius brevicaudus*	10/12	83.33	1-26
*Rhabdophis tigrinus*	132/162	81.48	1-28
*Elaphe schrenkii*	0/7	0	N/a
*Elaphe davidi*	0/2	0	N/a
*Agkistrodon saxatilis*	51/60	85.00	N/a
*Dendrelaphis pictus*	192/378	50.79	1-111
*Gloydius blomhoffi siniticus*	134/435	30.80	1-100
Total	1581/2934	53.89	1-294

**Table 3 animals-12-01216-t003:** Genetic diversity indices of *cox1* and *cyt*b genes in *Spiromerta* isolates from Hunan province, China, including sampling size (SS), number (n) of haplotypes, haplotype diversity (Hd), and nucleotide diversity (Pi).

mtDNA Gene	SS	n Haplotypes	Hd	Pi
*cox*1	67	30	0.899	0.0105
*cyt*b	67	26	0.888	0.0117
*cox*1 + *cyt*b	67	48	0.979	0.0110

**Table 4 animals-12-01216-t004:** Analysis of molecular variance (AMOVA) based on mtDNA sequences of the populations of *Spirometra* isolates.

Source of Variation	d.f	Sum of Squares	Variance Components	Percentage of Variation	Fixation Index (FST)	*p*-Value
Among populations	13	373.415	3.755210	24.96	0.24961	0.0000
Within populations	53	598.317	11.28899	75.04		
Total	66	971.731	15.04420	100		

**Table 5 animals-12-01216-t005:** Estimates of pairwise FST of concatenated sequences between sparganum populations.

Sample	1	2	3	4	5	6	7	8	9	10	11	12	13	14
1 YiY	0.00													
2 CD	−0.19	0.00												
3 YZ	−0.04	−0.03	0.00											
4 HY	0.24	0.27 *	0.29 *	0.00										
5 XT	0.17 *	0.17	−0.03	0.20	0.00									
6 SY	0.32	0.35	0.28	−0.00	0.14	0.00								
7 ZZ	0.43 *	0.47 *	0.43 *	0.07	0.32 *	−0.17	0.00							
8 CS	0.37 *	0.39	0.35	0.11	0.21	−0.13	−0.07 *	0.00						
9 LD	0.16 *	0.27	0.22	0.24	0.23	0.32	0.43 *	0.40 *	0.00					
10 CZ	−0.10	−0.04	−0.08	0.31	0.11	0.38	0.51	0.41 *	0.31	0.00				
11 HH	−0.24	−0.13	0.03	0.09	0.16	0.17	0.29	0.29 *	0.07	0.17	0.00			
12 ZJJ	0.30 *	0.33	0.17	−0.01	−0.08	−0.21	−0.05	−0.11	0.28	0.33	0.17	0.00		
13 YuY	0.04	0.05	0.09	0.36 *	0.18	0.42 *	0.54 *	0.44 *	0.26 *	0.11	0.10	0.40	0.00	
14 XX	−0.04	0.02	0.00	0.30	0.12	0.36	0.49 *	0.41 *	0.27	0.12	0.04	0.31	0.03	0.00

Significance of χ^2^: * *p*-value < 0.05.

## Data Availability

Data are contained within the article.

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
