# Peer review of "Epidemiology and Genetic Diversity of Spirometra Tapeworm Isolates from Snakes in Hunan Province, China"

_animals, 2022, doi:10.3390/ani12091216_

Round 1
Reviewer 1 Report
This manuscript describes the genetic diversity and epidemiology of Spirometra from snakes in the Hunan Province of China. This study is well designed and the information is useful and contributes to the scientific literature. There are a few items that should be addressed prior to accepting this manuscript for publication.
- Scientific publications should be written in third person. The authors will occasionally slip into first person language. Examples on lines: 16, 60-62, 92, 99, 150, 206, 210, 239, 258, 259, and 264. Please convert these statements into third person.
- Genus and species should be italicized, please correct. See lines: 42, 106, 138, 139, 141-144, 171, 172, 181, 189, and 207.
- Sentences should not begin with an Arabic number, please correct. See lines: 118, 236
- Rates are best expressed as a percentage followed by the number as a numerator and denominator. For example, line 28: (89.5%, 402/449). Please adjust for lines 28, 119-121, 125, 127-130, 232, and 275.
- Line 38: occurs
- Line 39, delete “There are” and capitalize “more”
- Lines 45-49 are awkward and confusing. Please re-write to clarify.
- Lines 55-56: “Spirometra decipiens” can be written “ decipiens”
- Line 71, do you mean “using techniques described by Liu et al., …”?
- Line 146-147. The method used to calculate intensity of infection is not described. Please correct table legend and heading. In Table 1 the authors list the intensity of infection as sparganum/snake. Lines 142-144 mention snakes with the highest intensity of 1-294, 1-291, and 1-208. Using the same method as Table 1, this would mean there was 1 sparganum per 294 snakes, etc. Please clarify.
- Line 233. “Snakes” should not be capitalized.
- Lines 238-239 may read better as “In the south and southwest regions of China, snakes are considered a delicacy.”
Author Response
Dear Reviewer 1:
Thank you for your suggestions. All your suggestions are very important, and they are of great guiding significance to my thesis writing and scientific research work.
According to your comments, we have made corresponding modifications in the article. Some of your questions will be answered below.
Point 1: Scientific publications should be written in third person. The authors will occasionally slip into first person language. Examples on lines: 16, 60-62, 92, 99, 150, 206, 210, 239, 258, 259, and 264. Please convert these statements into third person. Response 1: Thank you for your comments. We have converted these statements into the third person, and please see the article for details.
Point 2: Genus and species should be italicized, please correct. See lines: 42, 106, 138, 139, 141-144, 171, 172, 181, 189, and 207. Response 2: Thanks for your correction. We have made corresponding modifications in the article, and please see the article for details.
Point 3: Sentences should not begin with an Arabic number, please correct. See lines: 118, 236. Response 3: We have corrected the errors in these sentences. Please see in lines 118 and 239.
Point 4: Rates are best expressed as a percentage followed by the number as a numerator and denominator. For example, line 28: (89.5%, 402/449). Please adjust for lines 28, 119-121, 125, 127-130, 232, and 275. Response 4: Thanks for your correction. We have changed the rates to a percentage followed by the number as a numerator and denominator. Please see the article for details.
Point 5: Line 38: occurs and line 39, delete “There are” and capitalize “more” Response 5: We have changed these errors. Please see in lines 38 and 39.
Point 6: Lines 45-49 are awkward and confusing. Please rewrite to clarify.
Response 6: Thank you for your comments. We have rewritten it to clarify the meaning of this sentence. We want to show the inadequacy of studies on the prevalence of sparganum in snakes.
Point 7: Lines 55-56: “Spirometra decipiens” can be written “decipiens” Response 7: We have changed “Spirometra decipiens” with “S. decipiens”.
Point8: Line 71, do you mean “using techniques described by Liu et al., …”? Response 8: Thanks for your correction. We want to express “using the method described by Liu et al, 2020”. We have made corresponding modifications in the article.
Point 9: Line 146-147. The method used to calculate intensity of infection is not described. Please correct table legend and heading. In Table 1 the authors list the intensity of infection as sparganum/snake. Lines 142-144 mention snakes with the highest intensity of 1-294, 1-291, and 1-208. Using the same method as Table 1, this would mean there was 1 sparganum per 294 snakes, etc. Please clarify. Response 9: We have made corresponding modifications in the article. We want to show the range of numbers of spargana per snake. Thanks for your correction. Please see the article for details.
Point 10: Line 233. “Snakes” should not be capitalized. Response 10: Thanks for your correction. We have changed “Snakes” to “snakes”.
Point 11: Lines 238-239 may read better as “In the south and southwest regions of China, snakes are considered a delicacy.” Response 11: Thank you for your suggestions on this sentence. We have modified the corresponding position in the article. Please see the article for details.
Thank you again for your advice and I hope to learn more from you. If you have any further questions, please do not hesitate to contact me.
Yours sincerely,
Wei Liu

Reviewer 2 Report
In this manuscript, author investigated the epidemiology and genetic diversity of Spi-16 rometra tapeworms in snakes in the Hunan province. Result showed the higher infection rate about 89.53% compared to other regions. There was high genetic diversity among the parasitic isolates but no distinct genetic structure among 19 Spirometra populations. The phylogenetic analysis supported the division for European and Chinese 20 Spirometra isolates and the single species in Chinese Spirometra isolates. Manuscript is very well written and clearly concluded the outcomes of the study.
Minor comments:
Did authors have any information on the age of the snakes included in the study? Does age can affect the outcome of positive infection rate and prevalence of parasite in them? if yes, please mention about this in the manuscript.
Author Response
Dear Reviewer 2:
Thank you for your suggestions. Your suggestion is very important, and they are of great guiding significance to my thesis writing and scientific research work.
Point 1: Did authors have any information on the age of the snakes included in the study? Does age can affect the outcome of positive infection rate and prevalence of parasite in them? if yes, please mention about this in the manuscript.
Response 1: Thank you for your comments. We agree with you about that the age of the snakes can affect the outcome of positive infection rate and prevalence of parasite in them, and it will be great if we can record information on the age of the snakes in our study. Unfortunately, during the colletion of snakes, we can’t get the information about the accurate age of those wild snakes, for that it is hard to determine the age exactly under the current conditions.
Thank you again for your advice.
Yours sincerely,
Wei Liu

Reviewer 3 Report
Manuscript presents interesting data on the prevalence and genetic diversity of Spirometra cestode from snakes in the Hunan Province of China. The study is well designed. The Authors examined as many as 449 wild snakes. In my opinion the results are correctly presented and the manuscript is well written. However I have few minor suggestions for the Authors before publishing the article in Animals.
Line 24 (Abstract): please delete one “in wild snakes”
Line 25: Spirometra - please use the italics
Line 41: put a full stop at the end of the sentence
Line 42: Spirometra - please use the italics
Line 106: D. latum – please use the italics
Line 138-144: please use the italics when you mention the names of snake species
Line 149-154: “cytb” and “cox1” - please use the italics
Line 158: “cytb”, “cox1” and “Spirometra” please use italics
Line 171-172: please use the italics when you mention the names of host species
Line 181: “Spirometra” – use the italics
Line 207: “Spirometra” – use the italics
Line 232: “Spirometra” – use the italics
Line 248 and 270: “spp.” – please remove italics
Supplementary table S2 (Appendix B): replace “Marta et al., 2019” with “KoÅ‚odziej-SobociÅ„ska et al., 2019”
Author Response
Dear Reviewer 3:
Thank you for your suggestions. All your suggestions are very important, and they are of great guiding significance to my thesis writing and scientific research work.
According to your comments, we have made corresponding modifications in the article. Some of your questions will be answered below.
Point 1:Line 24 (Abstract): please delete one “in wild snakes”
Point 2: Line 25: Spirometra - please use the italics
Point 3: Line 41: put a full stop at the end of the sentence
Point 4: Line 42: Spirometra - please use the italics
Point 5: Line 106: D. latum – please use the italics
Point 6: Line 138-144: please use the italics when you mention the names of snake species
Point 7: Line 149-154: “cytb” and “cox1” - please use the italics
Point 8: Line 158: “cytb”, “cox1” and “Spirometra” please use italics
Point 9: Line 171-172: please use the italics when you mention the names of host species
Point 10: Line 181: “Spirometra” – use the italics
Point 11: Line 207: “Spirometra” – use the italics
Point 12: Line 232: “Spirometra” – use the italics
Point 13: Line 248 and 270: “spp.” – please remove italics
Point 14: Supplementary table S2 (Appendix B): replace “Marta et al., 2019” with “KoÅ‚odziej-SobociÅ„ska et al., 2019” Response: Special thanks to you for your good comments. According to your comments, all of the mistakes have been revised accordingly in the new MS. Please see the revised article for details.
As for line 248: Spirometra sp. 1, referring to the Spirometra study analyzed by Kuchta et al., 2021, which might a new species. After careful consideration, we reserve this amendment.
Thank you again for your advice.
Yours sincerely,
Wei Liu

Reviewer 4 Report
In the manuscript entitled “The epidemiology and genetic diversity of Spirometra tape worm isolates from snakes in the Hunan Province of China” the authors discuss the investigation of the prevalence of sparganum infection in wild snakes in the Hunan province. Below are the comments.
- Why did the authors choose to do genetic diversity analysis using only ctyb and cox1 genes? This should be explained in detail. The authors would have adopted the analysis of multiple genes to determine the genetic diversity.
- Previous investigation of the sparganum isolates from frogs posits that the Chinese Sprirometra population probably comprised of two sub-groups. However, the authors mention that there was no distinct structure of cytb and cox1 genes in the case of snakes in the Hunan province. How can these results be reconciled?
Author Response
Dear Reviewer 4:
Thank you for your suggestions. All your suggestions are very important, and they are of great guiding significance to my thesis writing and scientific research work.
According to your comments, we have made corresponding modifications in the article. Some of your questions will be answered below.
Point 1: Why did the authors choose to do genetic diversity analysis using only ctyb and cox1 genes? This should be explained in detail. The authors would have adopted the analysis of multiple genes to determine the genetic diversity.
Response 1: Thank you for your comments. In this study, we only choose two molecular markers: ctyb and cox1, to perform the genetic diversity analysis. The main reasons are as follows: 1) The ctyb and cox1 genes have been verified as suitable markers for inferring genetic population differences of Spirometra tapeworms (Zhang et al., 2015; Kolodziej-Sobocinska et al., 2019; Hong et al., 2020; Kuchta et al., 2021). 2) The other reason is that there are many published data of Spirometra ctyb and cox1 genes in the public database such as GenBank to compare previous studies. As the reviewer suggested, it will be better to perform the analysis using multiple genes. However, few available suitable markers can be used for other genes. Therefore, we performed the analysis just using two genetic markers.
Point 2: Previous investigation of the sparganum isolates from frogs posits that the Chinese Sprirometra population probably comprised of two sub-groups. However, the authors mention that there was no distinct structure of cytb and cox1 genes in the case of snakes in the Hunan province. How can these results be reconciled?
Response 2: Thank you for your good comments. We think there are probably two reasons: 1) The first one is the sampling size. Previous investigation of the sparganum isolates from frogs used more samples than this study. 2) The second possible reason could be the sampling range. Previous investigation of the sparganum isolates from frogs used samples from many different provinces, autonomous regions, and municipalities of China. In comparison, the geographic distances between populations studied here are relatively small.
Thank you again for your advice.
Yours sincerely,
Wei Liu
